# Model-Based Fault Diagnosis of an Anti-Lock Braking System via Structural Analysis

**DOI:** 10.3390/s18124468

**Published:** 2018-12-17

**Authors:** Qi Chen, Wenfeng Tian, Wuwei Chen, Qadeer Ahmed, Yanming Wu

**Affiliations:** 1School of Mechanical Engineering, Hefei University of Technology, Hefei 230009, China; hfgd_twf@126.com (W.T.); yanmingwu68@163.com (Y.W.); 2School of Automobile and Transportation Engineering, Hefei University of Technology, Hefei 230009, China; hfut_wwc@163.com; 3Center for Automotive Research, The Ohio State University, Columbus, OH 43212, USA

**Keywords:** fault detection and identification, anti-lock braking system, model-based, structural analysis, residual design

## Abstract

The anti-lock braking system (ABS) is an essential part in ensuring safe driving in vehicles. The Security of onboard safety systems is very important. In order to monitor the functions of ABS and avoid any malfunction, a model-based methodology with respect to structural analysis is employed in this paper to achieve an efficient fault detection and identification (FDI) system design. The analysis involves five essential steps of SA applied to ABS, which includes critical faults analysis, fault modelling, fault detectability analysis and fault isolability analysis, Minimal Structural Over-determined (MSO) sets selection, and MSO-based residual design. In terms of the four faults in the ABS, they are evaluated to be detectable through performing a structural representation and making the Dulmage-Mendelsohn decomposition with respect to the fault modelling, and then they are proved to be isolable based on the fault isolability matrix via SA. After that, four corresponding residuals are generated directly by a series of suggested equation combinations resulting from four MSO sets. The results generated by numerical simulations show that the proposed FDI system can detect and isolate all the injected faults, which is consistent with the theoretical analysis by SA, and also eventually validated by experimental testing on the vehicle (EcoCAR2) ABS.

## 1. Introduction

With the development of modern vehicle technologies, more and more advanced controlling systems related to driving comfort and safety are applied in the vehicles, such as the anti-lock braking system (ABS), electric brakeforce distribution (EBD), electronic stability program (ESP) or vehicle stability assist (VSA), and adaptive cruise control (ACC). It is true that these innovative configurations will improve the safety of the vehicles, but they will simultaneously add complexity to the controlling system. Once there is a malfunction in these systems, they will conversely result in safety accidents. Thus, real time fault detection, isolation, identification and tolerance for these systems are very vital to guarantee their normal operation and vehicle safety. This motivates the demand for fault detection and identification (FDI) of system and sensor faults for vehicles, which will be beneficial for the automotive companies to produce ISO 26262 compliant vehicles [1].

The literatures about the fault diagnosis for vehicles are tremendous, and can be classified into four categories—model-based, signal-based, knowledge-based or hybrid/active approaches according to Gao’s research [2,3]. With respect to the model-based approaches, the parity equations [4] and observers [5] are commonly used for their good adaption to both linear and non-linear systems. For example, parity relation based fault diagnosis are applied in automotive engines [6,7], the suspension system [8], and the motor-driven power steering system [9]. Literatures about fault diagnosis via observers [10] can be found used in vehicle lateral and yaw dynamics control systems [11,12], unmanned aerial vehicles [13,14], permanent-magnet synchronous motor (PMSM) [15,16], and lithium-ion battery pack in electric vehicles [17]. Moreover, these two methods are used simultaneously for complex systems, such as observer theory and parity space, and are both employed for fault diagnosis of an electric vehicles in reference [18], and for ABS fault diagnosis in reference [19].

With regard to signal-based methods, there is also much research on vehicles, such as fault diagnosis for PMSM [20] by digital signal processor, engine valve clearance by vibration signal [21], fault diagnosis of internal combustion engines [22] and automobile hydraulic brake system [23] by signals of vibration acceleration.

The knowledge-based methodology has been widely used in the diagnosis of vehicle systems, such as automotive engine [24,25], brake fault diagnosis [26], lithium-Ion battery packs of electric vehicles [27], and vehicle body [28].

Sometimes, more than one of the existing methods are integrated into fault diagnosis for complicated systems, which is called the “hybrid approach” [3]. For example, fault diagnoses of engines are analyzed in different combination of the above methods in references [29,30,31]; diagnosis and recovery for a team of unmanned vehicles are discussed in references [32,33], and detection and recognition of vehicles are studied in references [34].

Unlike the above fault diagnosis methods which are not invasive, active fault diagnosis method is an invasive method, which means that a designed signal can be inputted into the system under a test interval to quickly and accurately detect the faults [3]. Some investigations can be found in reference [35] for fault diagnosis of actuator and sensor of a vehicle, in reference [36] about the fault diagnosis of a quadrotor unmanned aerial vehicle, and in reference [37] for fault tolerant control of four-wheel independently driven electric ground vehicles.

Based on the above analysis, it is obvious that the topics around fault diagnosis for vehicles are still highly focused on because pursuit of more stable and safer vehicles never stops. It is well known that the anti-lock braking system (ABS) is a very important part of modern vehicles, which can effectively prevent side slip and tail flick, improve braking stability, reduce braking distance, and greatly enhance vehicle driving safety. Therefore, if there is malfunction in ABS, it will cause safety risks to the vehicle and even cause fatal accidents. For improving the reliability of ABS, fault detection and identification (FDI) for the ABS will be discussed in this paper.

At present, ABS fault diagnosis has been a focus for researchers due to its importance for the stability of braking systems in cars. Fault isolation for the hydraulic circuit of an ABS was earlier discussed by Sachenbacher and Struss et al. [38,39], and it was pointed out the techniques of qualitative modelling and failure mode and effects analysis (FMEA) are a key for fault diagnosis for the ABS.

Afterwards, more scholars have been engaged in the research of fault diagnosis for the ABS. Our investigation of the existing literatures about the ABS fault diagnosis has found a model-based methodology that has been utilized to fault diagnosis for the ABS by some of the authors. For example, a model-based fault diagnosis and isolation scheme was demonstrated by Pisu et al. [40] for automobile active braking systems, and a fault detection and isolation scheme design using parity equation based residual generation was presented on an ABS model. The model-based technology to fault diagnosis for automotive vehicle suspension and hydraulic brake systems was explained by Börner et al. [41], and a detailed process of fault detection of solenoid ABS valves using the parity equation method was shown to generate residuals for fault detecting. A sliding mode observer (SMO) was designed by Zahedi et al. [42] to detect and isolate the possible sensor faults in the ABS, and it was proved that their strategy is effective in isolating various types of sensor faults by numerical simulations. A sliding mode observer-based residual generator and gray-relational-analyzed strategy was also proposed by Dai et al. [43] for wheel sensor fault diagnosis in an aircraft ABS, and their diagnosis strategy was validated in four different sensor states. A mix of model-based and data-driven methods was presented by Luo et al. [19] to perform a fault detection and isolation system design and validation for ABS, and a fault detection and isolation scheme was carried out for fault detecting according to a subset of faults.

Some of the authors have tried to introduce modern controlling techniques like BP neural networks, petri nets or fuzzy theory into fault diagnosis for the ABS. BP neural networks in fault diagnosis for the ABS were utilized by Wang et al. [44], and the fault modes of actuators and sensors was built up, and the cases of system variables waving was investigated. Fault diagnosis analysis of ABS using petri nets was researched by Cabasino et al. [45], and the effectiveness of this method was validated for the ABS by performing a certain cases diagnosis. Their methodology is based on a logic reasoning, but did not depend on a mathematical model of the ABS, and they have not mentioned the fault detection and isolation issues. These two points are different from this paper. The fuzzy diagnosis technology for fault diagnosis on the ABS was employed by Ji et al. [46], where the fuzzy relation matrix used to reflect the correlation between fault cause and fault symptom was established, by which the conclusion of possible failure cause can be obtained for easier fault inspection.

There are also some scholars that have considered designing a multifunctional controlling system integrated with the function of fault diagnosis for the ABS. For instance, a dual-CPU structure based strategy was used by Yu et al. [47] to design a fault diagnostic system for an ABS, and the diagnostic system was validated by a road test of a left valve fault detecting in the pneumatic ABS. An integral-type sliding mode control (ISMC) scheme was engaged by Liang et al. [48] and was applied to the control system of the ABS, and it was proved that this scheme can also detect and tolerate the brake actuator fault.

From the above investigation of the existing literatures about ABS fault diagnosis, most of them have employed model-based technologies for health monitoring and fault detecting of ABS. In terms of the above model-based methodologies in ABS, there are three kinds of theory involved in these papers: parity equation [40,41], observer [42,43] or mixed parity equation and observer [19]. These model-based methodologies are now popular in the fault diagnosis on the ABS as well as the other vehicle systems [49,50,51,52,53,54,55]. However, none of them has done a systematical fault detectability (FD) or fault isolability (FI) analysis. (Here, FD means whether a fault can be detected when it occurs, and FI means whether a fault can be uniquely detected, that is, a fault can not only be detected, but also identified or distinguished from the other faults. These two indexes will reflect the safety and reliability of a control system). Moreover, the process of analyzing the fault detectability and isolability in the above methods is not intuitive; and the procedures of performing the FDI scheme for a residuals generator are not concise.

To resolve the above issues, another model-based methodology—structural analysis (SA) will be introduced in this paper. The theory of SA is derived from the conception of the bond graph [56]. Frisk and his team started earlier to study the theory of SA and have made great progress on the theoretical research of SA and its industrial application in complex system. For example, the SA theory was employed by Düştegör and Frisk [57] to analyze the fault isolability for the DAMADICS valve benchmark model, where it was proved that SA is an efficient way to rapidly evaluate the FI for a complicated system. The SA theory in a large and nonlinear model was also applied by Krysander and Frisk [58] to obtain an optimal sensor placement scheme for maximal capability of FD and FI, which is eventually validated in an industrial value. Recently, an updated MATLAB toolbox of SA was demonstrated by Frisk [59] to analyze the ability of FD and FI, and the diagnosis system for large scale models was designed, where another successful case of applying the methodology of SA in the air-path diagnosis of automotive engine was presented. Followed by Frisk’s research, the application of SA to design a diagnostic strategy was shown by Zhang [60,61] for an electric vehicle with a permanent magnet synchronous machine (PMSM); the SA method to implement a health monitoring scheme was also employed by us [62] for an automated manual transmission (AMT), where the FD and FI are easily obtained with available set of sensors, and then a robust FDI system is also efficiently designed by the SA method. Based on the above introduction of SA theory, it is known that SA is an effective method of fault diagnosis on mechanical and controlling systems, and it has an eminent advantage of easily performing detectability and isolability analysis and realizing an efficient FDI scheme design for a complex system.

Considering the above virtues of the SA theory and the successful applications of SA in linear [62] and nonlinear systems [57,59], it is believed that the SA method is also adapted to fault diagnosis and FDI scheme design for the ABS. For this reason, in this paper the efficient model-based fault diagnosis methodology of SA will also be employed to perform fault diagnosis for the ABS, which not only gives an optimal sensor location but also helps to shortlist the right number of residuals. Based on the successful cases of applying SA, it needs to know that when SA is executed; (1) a system with given set of sensors can be evaluated for the faults to be detected and isolated; (2) sequential residuals for FDI system design can be generated directly from the Minimal Structural Over-determined (MSO) sets from the SA theory. This paper will display the procedures thoroughly when applying SA into a fault diagnosis of ABS.

The rest of this paper is organized as follows. Section 2 is the main body of this paper and explains the detailed process of applying the SA method, where the critical faults in ABS are analyzed firstly; and then the fault model for ABS is established by the system model coupled with key faults; and then analysis of FD and FI with regard to the selected faults are obtained efficiently and intuitively by SA; finally, MSO sets are obtained and four residuals are designed. Section 3 presents the FDI system and its validation by numerical simulation. Section 4 shows the experimental validation on ECOCAR 2 from the Ohio State University, before our conclusions are summarized in Section 5.

## 2. Structure Analysis Based Fault Detection and Identification for ABS

Structure analysis (SA) is a kind of model-based technique, which also relies on the mathematical model of the system. Its virtue is to execute a quick and efficient analysis of fault detectability (FD) and fault isolability (FI) with respect to the possible faults in the system, realize a degree of FD and FI by executing a sensor placement, and obtain a practical design scheme of the FDI system. Figure 1 provides the main steps when applying SA. In the next section, a detailed illustration of fault diagnosis for ABS based on SA will be presented.

### 2.1. The Critical Faults Analysis for ABS

Anti-lock Braking System (ABS) is one of the key components in improving active safety of vehicles, which is composed of an electronic control unit, brake pressure regular and wheel angular speed sensor, et al. The structure of ABS is shown in Figure 2. When the car is braking on the roads with snow and ice, the braking force will exceed the adhesion capacity that the road can bear. As a consequence, the rolling wheels tend to slip on the road and even get locked. The function of ABS is to prevent occurrence of this phenomenon by controlling the brake force according to the estimation of the slip rate in real time by the wheel angular speed sensor and vehicle speed sensor.

Based on the functional analysis and related hazard reports [38,47,63] about ABS, there are several critical faults in the system, such as “the leakage of solenoid value”, “the fault of slip rate operator”, “internal error of electric control unit (ECU)”, “vehicle speed sensor fault”, and “wheel angular speed sensor fault”. Among those 5 faults, the first three faults—the leakage of solenoid value, the fault of slip rate operator, and ECU malfunction—are hard perceptual ones which may cause a serious hazard to ABS, even resulting in safety accidents. Table 1 shows the faults and their variable definitions, where we combine the fault of slip rate operator and internal error of ECU and denote as $f_{S}$ for simplicity, and also for the reason of that they are actually in the same unit.

### 2.2. Fault Modelling of ABS

Before the fault modeling of ABS, a simple ABS mathematical model [64] will be firstly introduced.

(1) Friction coefficient

During the braking process, the wheel will slip relative to the road surface, which will influence the friction coefficient between wheel and road. The test shows that the friction coefficient has a nonlinear relation with wheel slip, which is influenced by several factors, such as wheel slip, the type of the road surface, and environmental conditions like humidity, temperature, etc. Here the relationship of these two parameters is defined as follows.(1)μ=Γ(S)
where μ is the friction coefficient between wheel and road; S is the wheel slip which is defined as
(2)$$S=1-\frac{\omega_{w}}{\omega_{v}}$$
where S is the wheel slip; $\omega_{w}$ is the wheel angular speed; $\omega_{v}$ is the equivalent vehicle angular speed which is calculated by
(3)$$\omega_{v}=\frac{v_{v}}{R_{r}}$$
here, $v_{v}$ is the vehicle speed; $R_{r}$ is the wheel radius.

Usually, an empirical formula is used as the solution of the function—Γ. Figure 3 shows results of μ and S by statistical data.

(2) The friction torque of a single wheel

For sake of simplicity, a quarter of the vehicle is only considered, then the longitudinal friction of a single wheel is obtained by
(4)$$F_{f}=KNS=K\cdot \frac{mg}{4}\cdot \mu$$
where, $F_{f}$ is the wheel longitudinal friction, $K=\frac{u_{h}}{S_{T}}$, $u_{h}$ is the peak attachment coefficient. Set $u_{h}=0.2$; $S_{T}$ is desired wheel slip, let $S_{T}=0.2;N$ is the normal force of wheel to the ground, and is simplified as $\frac{mg}{4}$.

The friction torque of a single wheel is
(5)$$T_{f}=F_{f\cdot }R_{r}$$

(3) Braking torque

The braking torque is:(6)$$T_{b}=K_{f}\cdot F_{b}$$
where, $K_{f}$ is the braking coefficient depending on the area of the piston; $F_{b}$ is the braking force, calculated by:(7)$${\dot{F}}_{b}=\frac{100}{T_{B}\cdot s+1}\mathrm{SIGN}\left(S_{T}-S\right)$$
here, $\frac{100}{T_{B}\cdot s+1}$ reflects the hydraulic system, where we assume it is a linear system; $T_{B}$ is time constant, set as 0.01; SIGN$\left(S_{T}-S\right)$ is a bang-bang type controller affected by the input variable $S_{T}-S$.

(4) Vehicle model

The total torque of the 1/4 vehicle model is
(8)$$T=T_{f}-T_{b}=K\cdot \frac{mg}{4}\cdot \mu \left(S\right)\cdot R_{r}-K_{f}F_{b}$$

According to the wheel motion equation, the wheel angular velocity can be obtained as follows:(9)$$T=I\cdot {\dot{\omega }}_{w}$$
where, I is the wheel inertia.

(5) Vehicle speed

The wheel is driven by longitudinal friction of wheel, and they have the following relation during the braking process.
(10)$$-\frac{F_{f}}{m\cdot R_{r}}={\dot{\omega }}_{v}$$

Combined with the above modeling, and “the critical faults of ABS “ in Section 2.1, the fault model of ABS system can be obtained as,
(11)$$\left\{ \begin{array}{l} e_{1}:T=K\cdot \frac{mg}{4}\cdot \mu \cdot R_{r}-f_{Kf}\cdot K_{f}\cdot F_{b}\\ e_{2}:{\dot{F}}_{b}=\frac{100}{TB\cdot s+1}\cdot \text{SIGN}\left(S_{T}-S\right)\\ e_{3}:T=I\cdot {\dot{\omega }}_{w}\\ e_{4}:-\frac{K\cdot g\cdot \mu }{4}={\dot{v}}_{v}\\ e_{5}:\omega_{v}=\frac{v_{v}}{R_{r}}\\ e_{6}:S=\left(1-\frac{\omega_{w}}{\omega_{v}}\right)\\ e_{7}:\mu =\Gamma \left(S\cdot f_{S}\right)\\ e_{8}:y_{v_{v}}=v_{v}+f_{v_{v}}\\ e_{9}:y_{\omega_{w}}=\omega_{w}+f_{\omega_{w}} \end{array}\right.$$
where $f_{kf}$ is the leakage fault of solenoid valve; $f_{S}$ is the slip-ratio operation fault. $e_{i}$ represents equation *i*; $f_{v_{v}}$ is the vehicle speed sensor fault; $f_{\omega_{w}}$ is the wheel angular speed sensor fault. $y_{v_{v}}$ and $y_{\omega_{w}}$ are the measurement of $v_{v}$ and $\omega_{w}$.

*Note*: here it is assumed that the fault of leakage of solenoid value and slip-ratio operation are gain type; the sensor fault of vehicle speed and wheel angular speed are bias type.

### 2.3. Fault Detectability Analysis and Fault Isolability Analysis by SA

The detectability and isolability mean that a fault can be identified and located when it occurs. In this section, by employing the technique of Dulmage-Mendelosohn (DM) decomposition in SA, the fault detectability (FD) analysis and fault isolability (FI) analysis of ABS will be discussed with the above four critical faults, and explored whether all the 4 faults are detectable and isolable.

(1) System structure representation

Structural representation is an approach to visually display the structure of the model [65], which is an important step in SA. Firstly, the variables in fault model at (11) will be divided into three groups: unknown variables, known variables and fault variables. Then the structure representation diagram is shown in Figure 4, where the unknown variables are $\left\{T,\mu ,F_{b},\omega_{w},v_{v},\omega_{v},S\right\}$, known variables are $\left\{S_{T};y_{v_{v}};y_{\omega_{w}}\right\}$, and the fault variables are $\left\{f_{Kf};f_{S};f_{v_{v}};f_{\omega_{w}}\right\}$.

Here, the symbol “×” indicates that the variable corresponding to the abscissa appears in the equation of the ordinate.

(2) Fault detectability (FD) analysis

In the methodology of SA [58], DM decomposition can be used to achieve fault detectability analysis intuitively. a DM decomposition can be executed by the command—“dmperm”—in MATLAB. By DM decomposition [66], the equations will be rearranged into three parts: under-determined part $M^{-}$, just-determined part $M^{0}$ and over-determined part $M^{+}$, as illustrated in Figure 5.

Here, the underdetermined part $M^{-}$, the just determined part $M^{0}$ and the overdetermined part $M^{+}$ means that the number of equations is less than, equal to, and more than unknown variables in the equations respectively. According to the theory of SA, if the fault lies in $M^{-}$ or $M^{0}$, the fault is undetectable because there is no redundant equation there. If the fault lies in $M^{+}$, the fault is detectable as there is an additional equation which can generate a redundant relation.

Figure 6a shows results of DM decomposition for the ABS, in which it can be seen that the faults ($f_{Kf};f_{S};f_{v_{v}};f_{\omega_{w}}$) are all in the $M^{+}$ area, so they are detectable.

(3) Fault isolability analysis

It is pointed out by Frisk et al. [58] that if a fault $f_{i}$ is isolable from $f_{j}$, this means that in the absence of equation $e_{fj}$, the equation $e_{fi}$ is still in the structural overdetermined area, and can be designed as:(12)$$e_{fi}\;\in {\left(M/e_{f_{j}}\right)}^{+}$$
where $e_{f_{i}}$, $e_{f_{j}}$ represent equations with faults $f_{i}$ and $f_{j}$ respectively, and M denotes the system model, ${\left(M/e_{f_{j}}\right)}^{+}$ indicates that removing the equation with fault $f_{j}$, the model is still overdetermined, in brief, the isolable fault means that the fault $e_{f_{i}}$ is only related to itself in the two-dimensional auto correlation matrix which is called the fault isolability matrix (FIM) here.

Based on the above definition, the FIM of ABS is indicated in Figure 6b, where it can be seen that the fault is only related to itself, hence all the four faults in the ABS can be isolated. According to the above analysis, it can be concluded that the four critical faults in the ABS system are not only detectable but also isolable.

*Note*: Armed with the two sensors of speed sensor and wheel angular speed sensor, all the 4 faults are detectable and isolable, so here the sensor placement analysis is not executed. The sensor placement tool to reach maximal FDI ability will be used if there is a fault that is neither detectable nor isolable. Please find the application in reference [58,67].

### 2.4. Finding Minimal Structural Over-Determined(MSO) Sets

MSO sets is an important step prior to the residual design when SA is executed. Here a MSO-algorism from Krysander [58] is also employed. The algorithm is based on a top-down approach in the sense that the entire model is started and then the size of the model is reduced step by step until a MSO set remains. By employing the algorithm in literature [58], the available MSO sets can be obtained by programming listed in Table 2, where the equations related to every MSO set are also presented.

Here, the symbol “×” indicates that the fault is undetectable, and “●” denotes that the fault can be detected in a MSO set.

### 2.5. Residual Design for FDI System

According to the above four MSO sets, 4 residuals can be generated correspondingly.

(1) Residual 1

Residual-1 can be obtained by the MSO1, where 6 equations are involved. They are $\left\{e_{4},e_{5},e_{6},e_{7,}e_{8},e_{9}\right\}$.
(13)$$\{\begin{array}{l} e_{4}:-\frac{K\cdot g\cdot \mu }{4}={\dot{v}}_{v}\\ e_{5}:\omega_{v}=\frac{v_{v}}{R_{r}}\\ e_{6}:S=\left(1-\frac{\omega_{w}}{\omega_{v}}\right)\\ e_{7}:\mu =\Gamma \left(S\right)\\ e_{8}:y_{v_{v}}=v_{v}\\ e_{9}:y_{\omega_{w}}=\omega_{w} \end{array}$$

According to the function of MSO sets [68], the formulas in (13) can generate a series of residuals, but some of them may have derivative elements which are unwelcome in a control system. To avoid a derivative part in the residual, equations of $e_{5},e_{6},e_{7},e_{9}$ are firstly substituted into $e_{4}$, then the following equation is obtained:(14)$$-\frac{K\cdot g}{4}\left(1-\frac{y_{\omega_{w}}}{y_{v_{v}}/R_{r}}\right)={\dot{y}}_{v_{v}}$$

It is almost impossible to get a direct mathematical expression of the solution-$\omega_{v}$ by (14), so here a fourth-fifth order Runge-Kutta numerical integration algorithm is utilized to resolve (14), which is relatively easily implemented in MATALB. Here we set the answer of $y_{v_{v}}$ as ${\hat{y}}_{v_{v}}$. Then the residual-1 is shown as follows.
(15)$$R_{1}=y_{v_{v}}-{\hat{y}}_{v_{v}}$$

(2) Residual 2

Residual-2 can be obtained by the MSO2, where 8 equations are included. They are $\left\{e_{1},e_{2},e_{3},e_{4},e_{5},e_{6},e_{8},e_{9}\right\}$.
(16)$$\left\{ \begin{array}{l} e_{1}:T=K\cdot \frac{mg}{4}\cdot \mu \cdot R_{r}-K_{f}\cdot F_{b}\\ e_{2}:\dot{F_{b}}=\frac{100}{TB\cdot s+1}\cdot \text{SIGN}\left(S_{T}-S\right)\\ e_{3}:T=I\cdot {\dot{\omega }}_{w}\\ e_{4}:-\frac{K\cdot g\cdot \mu }{4}={\dot{v}}_{v}\\ e_{5}:\omega_{v}=\frac{v_{v}}{R_{r}}\\ e_{6}:S=\left(1-\frac{\omega_{w}}{\omega_{v}}\right)\\ e_{8}:y_{v_{v}}=v_{v}\\ e_{9}:y_{\omega_{w}}=\omega_{w} \end{array}\right.$$

To generate a robust residual not including the derivative part, here the strategy in literature [69] by an analytic redundant relation (ARR) is also employed. After substituting all the equations ($e_{2}–e_{6}$,$e_{8}$,$e_{9}$) into $e_{1}$, an ARR is shown as:(17)$$I\cdot {\dot{y}}_{\omega_{w}}+mR_{r}\cdot {\dot{y}}_{v_{v}}+K_{f}\cdot F_{b}=0$$

This ARR can generate residual-2 in state-space form given by,
(18)$$\{\begin{array}{l} \dot{x}=-\beta_{2}(x+I\cdot y_{\omega_{w}}+mR_{r}\cdot y_{v_{v}})+K_{f}F_{b}\\ r_{2}=x+I\cdot y_{\omega_{w}}+mR_{r}\cdot y_{v_{v}} \end{array}$$
where, $\beta_{2}$ should be more than 0 for the stability of the system, and, and the same of requirements with respect to the following variables—$\beta_{3},$ and $\beta_{4}$.

(3) Residual 3

Residual-3 can be acquired by the MSO3, where 8 equations are related. They are $\left\{e_{1},e_{2},e_{3},e_{4},e_{5},e_{6},e_{7},e_{9}\right\}$.
(19)$$\left\{ \begin{array}{l} e_{1}:T=K\cdot \frac{mg}{4}\cdot \mu \cdot R_{r}-K_{f}\cdot F_{b}\\ e_{2}:\dot{F_{b}}=\frac{100}{TB\cdot s+1}\cdot \text{SIGN}\left(S_{T}-S\right)\\ e_{3}:T=I\cdot {\dot{\omega }}_{w}\\ e_{4}:-\frac{K\cdot g\cdot \mu }{4}={\dot{v}}_{v}\\ e_{5}:\omega_{v}=\frac{v_{v}}{R_{r}}\\ e_{6}:S=\left(1-\frac{\omega_{w}}{\omega_{v}}\right)\\ e_{7}:\mu =\Gamma \left(S\right)\\ e_{9}:y_{\omega_{w}}=\omega_{w} \end{array}\right.$$

By the same method of generating residual-1, the identical equation with (14) is also firstly setup, and then the solution of $v_{v}$ is obtained, which is also defined as ${\hat{y}}_{v_{v}}$. Then after substituting $e_{2}$, $e_{3}$, $e_{4}$ to $e_{1}$, an ARR is obtained by
(20)$$I\cdot {\dot{y}}_{\omega_{w}}+mR_{r}\cdot {\dot{\hat{y}}}_{v_{v}}+K_{f}\cdot F_{b}=0$$

This ARR can generate residual-3 in state-space form given by,
(21)$$\{\begin{array}{l} \dot{x}=-\beta_{3}(x+I\cdot y_{\omega_{w}}+mR_{r}\cdot {\hat{y}}_{v_{v}})+K_{f}F_{b}\\ r_{3}=x+I\cdot y_{\omega_{w}}+mR_{r}\cdot {\hat{y}}_{v_{v}} \end{array}$$

(4) Residual 4

Residual-4 can be attained by the MSO4, where 8 equations are associated. They are $\left\{e_{1},e_{2},e_{3},e_{4},e_{5},e_{6},e_{7},e_{8}\right\}$.
(22)$$\left\{ \begin{array}{l} e_{1}:T=K\cdot \frac{mg}{4}\cdot \mu \cdot R_{r}-K_{f}\cdot F_{b}\\ e_{2}:\dot{F_{b}}=\frac{100}{TB\cdot s+1}\cdot \text{SIGN}\left(S_{T}-S\right)\\ e_{3}:T=I\cdot {\dot{\omega }}_{w}\\ e_{4}:-\frac{K\cdot g\cdot \mu }{4}={\dot{v}}_{v}\\ e_{5}:\omega_{v}=\frac{v_{v}}{R_{r}}\\ e_{6}:S=\left(1-\frac{\omega_{w}}{\omega_{v}}\right)\\ e_{7}:\mu =\Gamma \left(S\right)\\ e_{8}:y_{v_{v}}=v_{v} \end{array}\right.$$

Similar to the procedures in residual-2 and residual-3, equations of $e_{2},e_{3},e_{5},e_{6},e_{7},e_{8}$ are firstly substituted into $e_{1}$, then the following equation is obtained by
(23)$$K\cdot \frac{mg}{4}\cdot \Gamma \left(1-\frac{\omega_{w}}{y_{v_{v}}/R_{r}}\right)\cdot R_{r}-K_{f}\cdot F_{b}=\;I\cdot {\dot{\omega }}_{w}$$

By utilizing a fourth-fifth order Runge-Kutta numerical integration algorithm to resolve (23), the answer of $\omega_{w}$ is obtained, denoted as ${\hat{y}}_{\omega_{w}}$. Then the $e_{1}$ as an ARR is given by
(24)$$I\cdot {\dot{\hat{y}}}_{\omega_{w}}+mR_{r}\cdot {\dot{y}}_{v_{v}}+K_{f}\cdot F_{b}=0$$

So residual-4 can be expressed in a state-space form, as
(25)$$\{\begin{array}{l} \dot{x}=-\beta_{4}(x+I\cdot {\hat{y}}_{\omega_{w}}+mR_{r}\cdot y_{v_{v}})+K_{f}F_{b}\\ r_{4}=x+I\cdot {\hat{y}}_{\omega_{w}}+mR_{r}\cdot y_{v_{v}} \end{array}$$

## 3. Design and Validation of FDI System

### 3.1. FDI System Design

In order to test the proposed 4 residuals, in this section the FDI system of ABS will be established and then every residual is evaluated after injecting several faults in the system. Here the FDI system is just designed based on the above 4 residuals.

Figure 7 shows the structure of the FDI system, where all the residuals are compared with a threshold value; and if it compasses it, there will be a fault; otherwise it will be a healthy state. Here, a fixed number as the threshold [70] is also used.

In order to examine the FDI system, 4 different faults are injected in the ABS system model. Table 3 displays the assumed faults type and occur time.

Figure 8 and Figure 9 gives a simulation of system response when the two system faults—$f_{kf}$ and $f_{S}$—happen. Figure 8 indicates that the fault $f_{Kf}$ occurs at 1.5–3.5 s, the equivalent vehicle angular speed and wheel angular speed both rises relative to the healthy state; the stopping distance for hard braking also increases a little. Figure 9 shows that when the fault $f_{S}$ occurs at 5–7 s, the equivalent vehicle angular speed increases but the wheel angular speed drops firstly and then fluctuates up and down; the stopping distance rises a little bit in the end.

Here, the related parameter to the simulation model are set as follows. K=1, m=912 kg, $R_{r}=0.3\;m$, $K_{f}=1$, $T_{B}=0.01$, $S_{T}=0.2$, $I=0.21\mathrm{kg}\cdot m^{2}$, $v_{0}=$ 26.8 m·s^−1^, $\beta_{2}=\beta_{3}=\beta_{4}=1$.

From the above analysis, it is necessary to know when there is a system fault in ABS, although the vehicle and wheel angular speed changes, the stopping distance varies by an amount that is not obvious, so it is difficult for drivers to perceive it. Therefore, it is necessary to give a quick and accurate fault diagnosis on the faults when developing an FDI system.

In next section, an FDI system based on the above designed 4 residuals will be designed to detect and locate the faults.

### 3.2. Residual Testing

Based on the model in (11) and the residual equations from (13) to (25), a corresponding MATALB model can be implemented in the MATLAB Simulink tool.

After running the simulation with injected faults, the responses of the above 4 faults are shown in Figure 10, Figure 11, Figure 12 and Figure 13.

As can be seen from Figure 10, Figure 11, Figure 12 and Figure 13, residual 1 can detect the faults-$f_{S},\;f_{v_{v}}$ and $f_{\omega_{w}}$ except for fault-$f_{Kf}$; residual 2 can detect the fault-$f_{Kf},\;f_{v_{v}}$ and $f_{\omega_{w}}$ except for the fault-$f_{S}$; residual 3 can detect the fault-$f_{Kf},\;f_{S}$ and $f_{\omega_{w}}$ except for the fault-$f_{v_{v}}$; residual 4 can detect the fault-$f_{Kf},\;f_{S}$ and $f_{v_{v}}$ except for the fault-$f_{\omega_{w}}$. Table 4 gives the summary of the residual results, in which we can see that the result of FDI by residuals is consistent with that of theoretical analysis in Table 2. Thus, the proposed FDI scheme by SA is feasible.

Here, the symbol “×” indicates that the fault is undetectable, and “●” denotes that the fault can be detected.

## 4. Experimental Validation

To further validate the proposed FDI system for ABS, here an off-line approach of fault injection is employed for the preliminary testing. The reason why we do not implement the on-line testing is that it may be fatal for the driver. The car here to be discussed is the EcoCAR2, which is a Chevrolet Malibu and has been modified into a hybrid electric car for the competition.

### 4.1. Experimental Set-Up

Figure 14 displays the whole structure of the experimental set-up. The left side is the powertrain structure of the testing car including the ABS in the wheels, and the ABS system in the rear wheels will be studied. The power there is an electric machine, which will transfer the torque and speed to the differential, and then to the wheels via a single-speed transmission. The wheel angular speed $\omega_{w}$ can be obtained by the electric machine speed indirectly. The vehicle speed $v_{v}$ can be obtained by the speed sensor, which is installed in the differential. First, the dSPACE Autobox is installed in the car in charge of collecting the signals of the wheel angular speed sensor and vehicle speed sensor, which have already installed in the vehicle. Second, the measurement of these two speed sensors is extracted and imported into the same FDI system, which has already obtained in the simulation part. Then, two faults into the vehicle speed sensor and wheel angular speed sensor are injected respectively. Finally, after setting the specific parameters under the experimental environment, the FDI system is run in MATLAB Simulink to examine the response of the four residuals, and then whether the residuals can detect the injected faults is observed, and finally whether the faults can be isolated is also judged.

### 4.2. FDI System Testing

Figure 15a shows results of the equivalent vehicle angular speed and wheel angular speed, in which their units are unified as rad per second. During the driving cycle, a state of a braking process which lies in 315 s–328.7 s is picked up. Figure 15b displays the details of equivalent vehicle angular speed and wheel angular speed.

Because an off-line experiment is being executed, only the sensor faults can be observed. Here it is assumed the vehicle speed sensor is a bias type, and they have the same injected faults with the simulation condition, which are displayed in Table 5. Figure 16 presents the wheel angular speed and vehicle speed sensor signals with the injected faults corresponding to Table 5. Then, after entering the parameters of EcoCAR2 [71,72] shown in Table 6 and running the FDI system in the simulation system, the responses of the 4 residuals can be obtained with respect to the injected two sensor faults.

Figure 17 shows results of the four residuals when no fault is present. From Figure 18 and Figure 19, we can find that when speed sensor fault appears, all the residuals except residual-1 can detect the fault, and when wheel angular speed sensor fault occurs, only residual-4 cannot detect it. The result is consistent with the theoretical result in Table 2 and Table 4, so the proposed FDI system is correct and practical.

## 5. Conclusions

Fault diagnosis for ABS based on SA has been performed in this paper, where the methodology of SA is proved again to be an effective way to assess the fault detectability and design of the FDI system.

In terms of the ABS system mentioned above, firstly four vital fault candidates are concluded to be detectable and isolable by the technique of DM decomposition in SA that is presently intuitively in the graphical form; related illustrations are given in Figure 4 and Figure 6. Another key contribution of SA is helpful for efficiently realizing the FDI system design. In case of the ABS here, four MSO sets are easily obtained by the algorithm from SA, which can be directly used for generating sequential residuals. By employing MATLAB/Simulink tools, the residual responses have been investigated with respect to the injected faults. The numerical results show that the FDI system can detect and completely isolate all four faults, which is also validated by off-line experimental testing on the EcoCAR2 from the Ohio State University that is clearly shown in Figure 18 and Figure 19.

In the future, the on-line semi-physical simulation and experimental validation will be studied with regard to the FDI system in this paper.

## Figures and Tables

**Figure 1 sensors-18-04468-f001:**
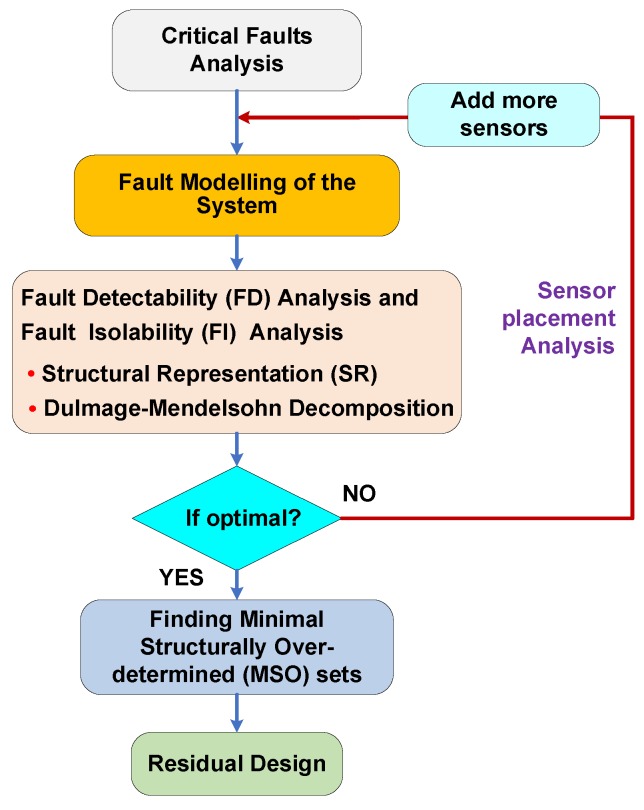
Main steps of structural analysis (SA).

**Figure 2 sensors-18-04468-f002:**
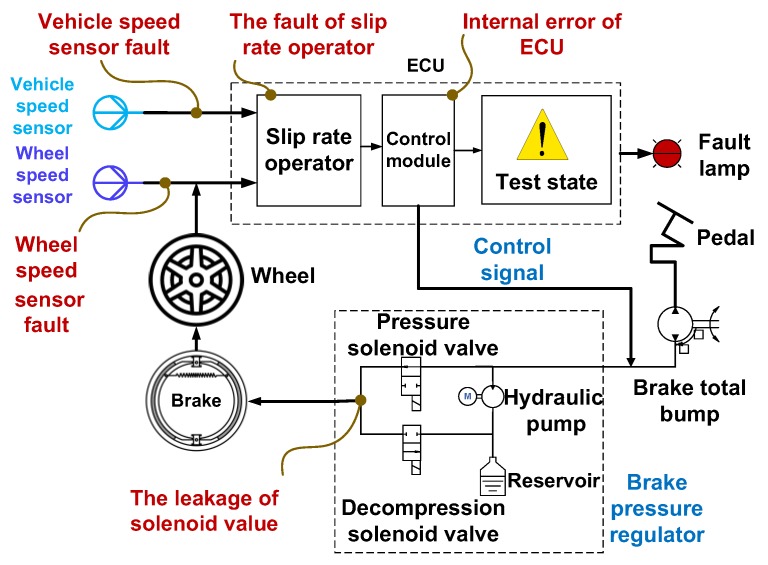
The structure diagram of the ABS system.

**Figure 3 sensors-18-04468-f003:**
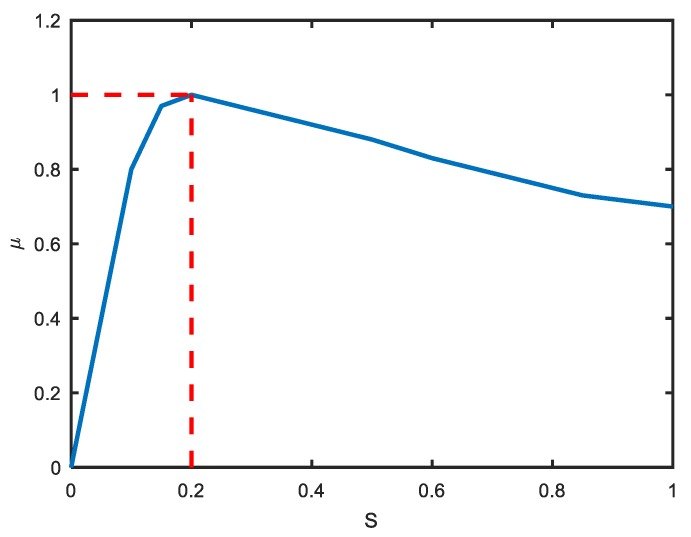
The relation between friction coefficient and wheel slip, where μ is the friction coefficient between wheel and road; S is the wheel slip. *Note*: here we just show one case of dry road.

**Figure 4 sensors-18-04468-f004:**
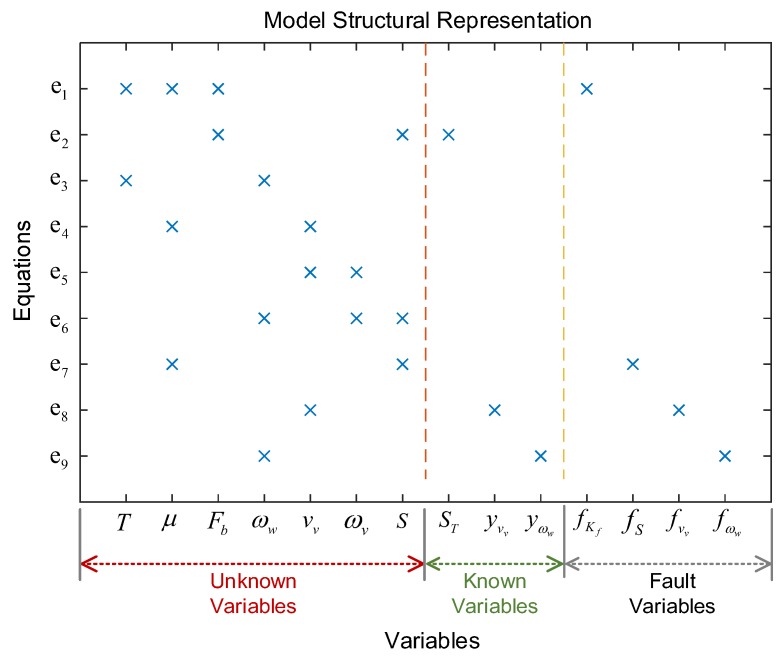
The structure representation diagram of system.

**Figure 5 sensors-18-04468-f005:**
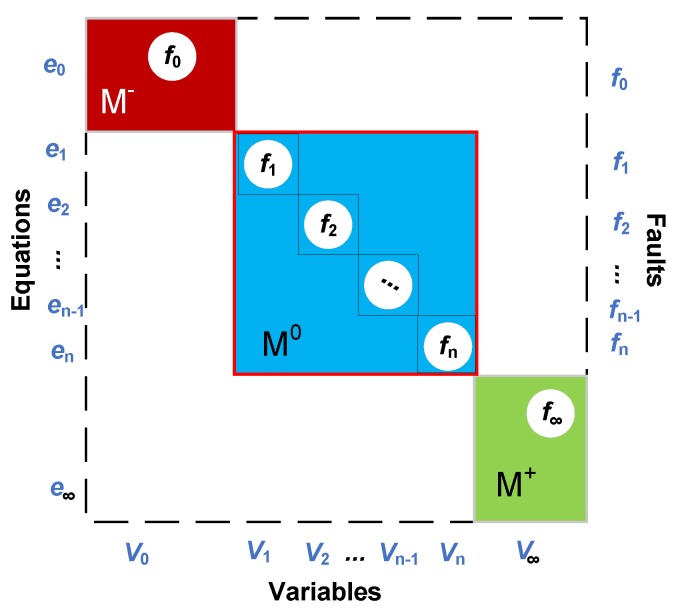
The schematic diagram of DM decomposition. where, $e_{0}-e_{\infty }$ represents the equations; $V_{0}-V_{\infty }$ represents the system variables; $f_{0}-f_{\infty }$ represents the fault variables.

**Figure 6 sensors-18-04468-f006:**
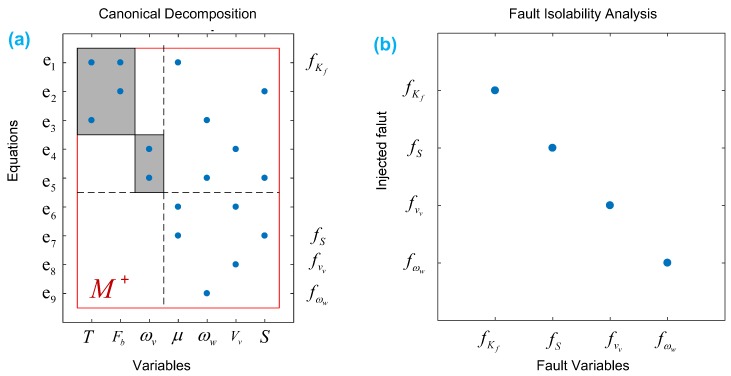
Analysis of fault detectability and isolability of initial ABS system. (**a**) DM-decomposition of the ABS (**b**) fault isolability matrix (FIM) of the ABS.

**Figure 7 sensors-18-04468-f007:**
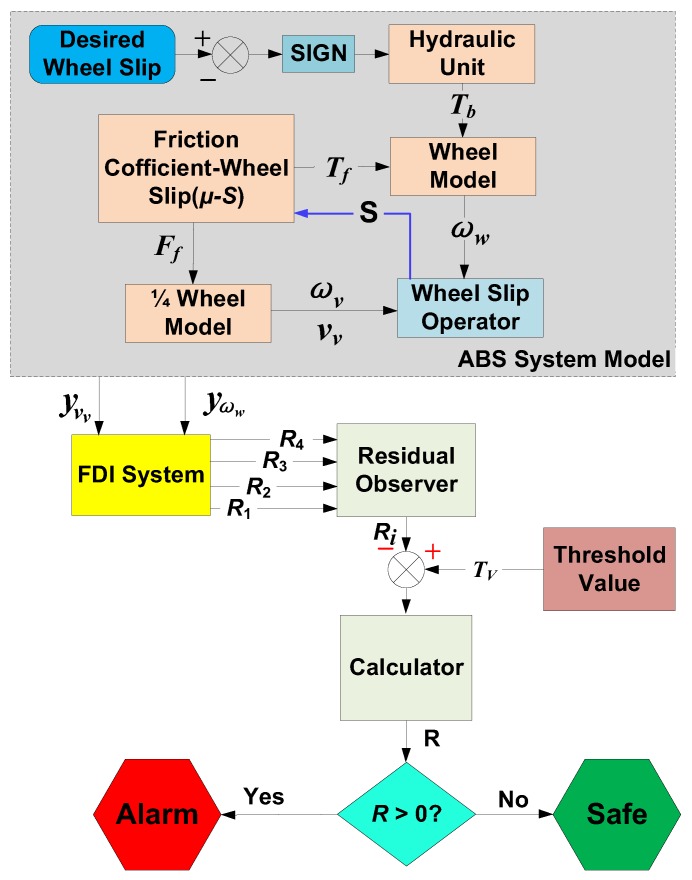
The schematic structure of FDI system.

**Figure 8 sensors-18-04468-f008:**
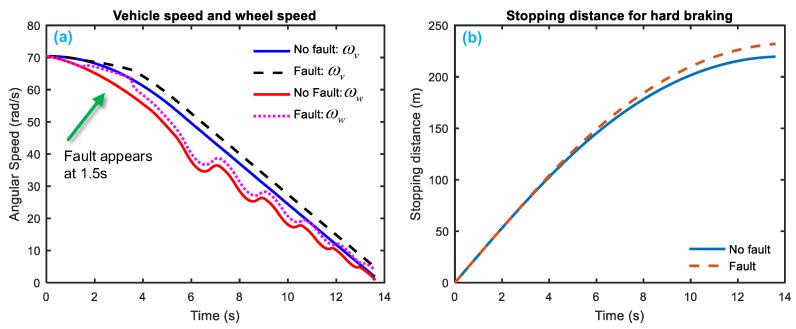
System response with faults of $f_{Kf}$. (**a**) speed reaction (**b**) stopping distance reaction.

**Figure 9 sensors-18-04468-f009:**
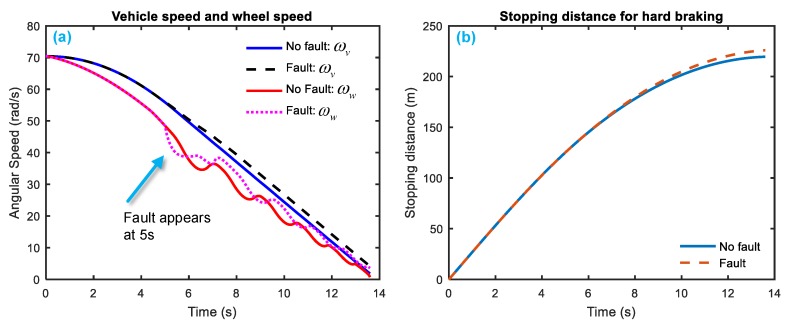
System response with faults of $f_{S}$. (**a**) speed reaction (**b**) stopping distance reaction.

**Figure 10 sensors-18-04468-f010:**
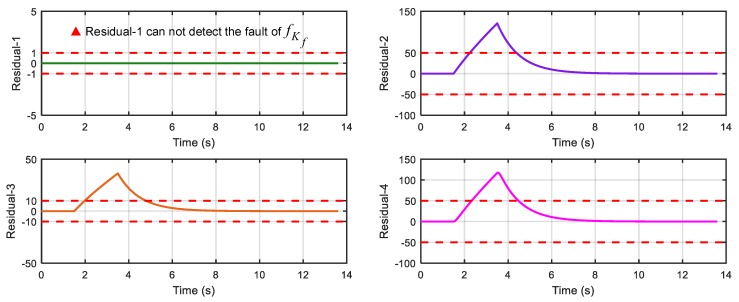
Response of 4 residuals with fault-$f_{Kf}$.

**Figure 11 sensors-18-04468-f011:**
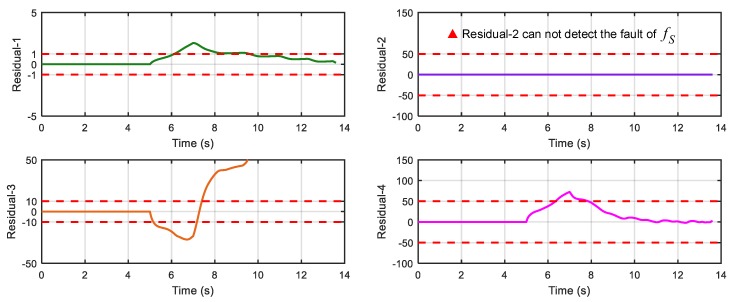
Response of 4 residuals with fault-$f_{S}.$

**Figure 12 sensors-18-04468-f012:**
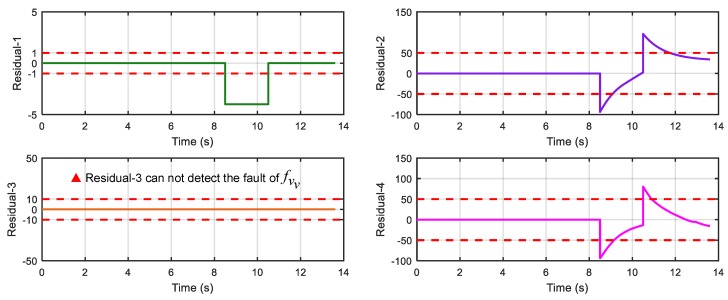
Response of 4 residuals with fault-$f_{v_{v}}$.

**Figure 13 sensors-18-04468-f013:**
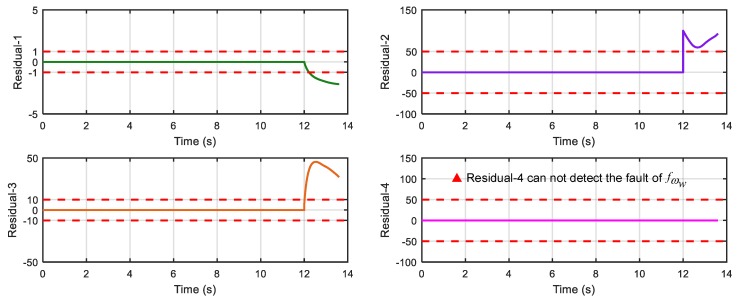
Response of 4 residuals with fault-$f_{\omega_{w}}$.

**Figure 14 sensors-18-04468-f014:**
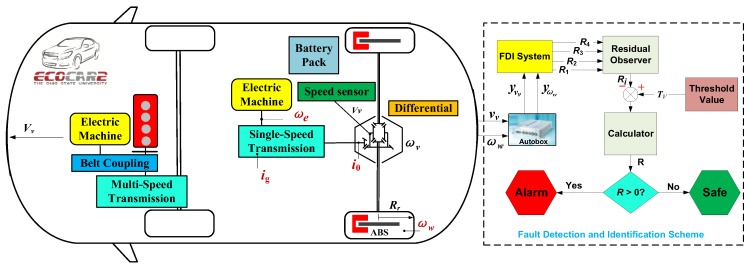
The experimental set-up of FDI scheme in a vehicle of EcoCAR2 from the OSU.

**Figure 15 sensors-18-04468-f015:**
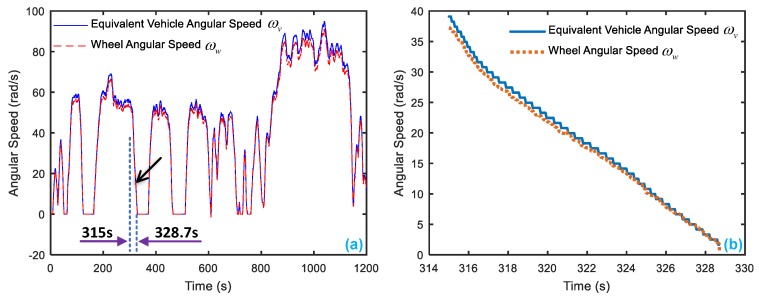
The angular speed of vehicle and wheel in a driving cycle, where $\omega_{v}$ is the equivalent vehicle angular speed, and $\omega_{w}$ is the wheel angular speed. (**a**) the whole driving cycle (**b**) selected section in a braking process.

**Figure 16 sensors-18-04468-f016:**
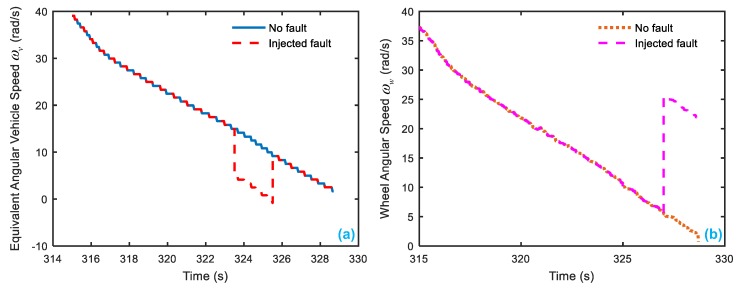
The speed sensor signal with injected fault. (**a**) the equivalent angular speed sensor fault with injected a bias fault at 323.5–325.5 s; (**b**) the wheel angular speed sensor fault with injected a bias fault at 327–329 s.

**Figure 17 sensors-18-04468-f017:**
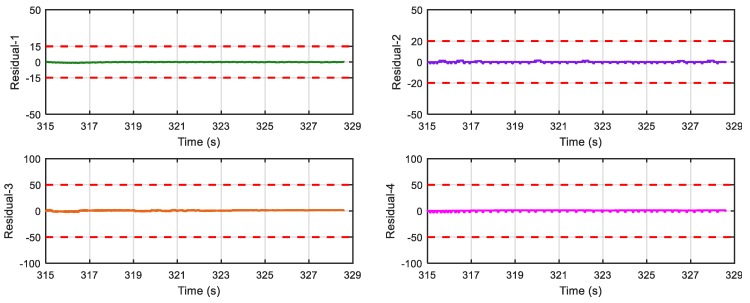
Response of the 4 residuals when no fault is present.

**Figure 18 sensors-18-04468-f018:**
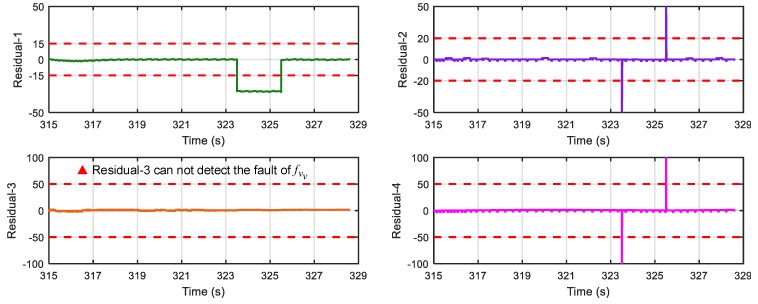
Response of the 4 residuals when speed sensor has fault ($f_{v_{v}}$).

**Figure 19 sensors-18-04468-f019:**
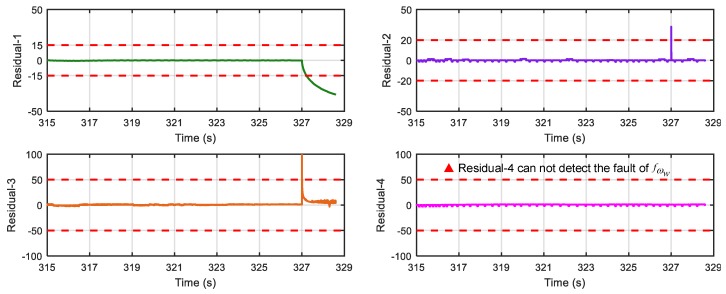
Response of the 4 residuals when wheel angular speed sensor has fault ($f_{\omega_{w}}$).

**Table 1 sensors-18-04468-t001:** Critical faults of ABS.

Faults	Faults Variables
Leakage of solenoid value	$f_{kf}$
The fault of slip rate operator/Internal error of ECU	$f_{S}$
Vehicle speed sensor fault	$f_{v_{v}}$
Wheel angular speed sensor fault	$f_{\omega_{w}}$

**Table 2 sensors-18-04468-t002:** MSO sets and corresponding equations.

	$f_{kf}$	$f_{S}$	$f_{v_{v}}$	$f_{\omega_{w}}$	Equations
${\mathrm{MSO}}_{1}$	×	●	●	●	$e_{4},e_{5},e_{6},e_{7,}e_{8},e_{9}$
${\mathrm{MSO}}_{2}$	●	×	●	●	$e_{1},e_{2},e_{3},e_{4},e_{5},e_{6},e_{8},e_{9}$
${\mathrm{MSO}}_{3}$	●	●	×	●	$e_{1},e_{2},e_{3},e_{4},e_{5},e_{6},e_{7},e_{9}$
${\mathrm{MSO}}_{4}$	●	●	●	×	$e_{1},e_{2},e_{3},e_{4},e_{5},e_{6},e_{7},e_{8}$

**Table 3 sensors-18-04468-t003:** Fault Setting Details.

Fault	Type	Signal	Time Span
$f_{Kf}$	Gain (0.95)	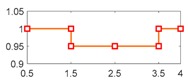	1.5–3.5 s
$f_{S}$	Gain (0.5)	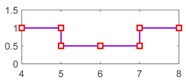	5–7 s
$f_{v_{v}}$	Bias (−10)	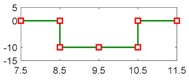	8.5–10.5 s
$f_{\omega_{w}}$	Bias (+20)	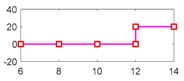	12–14 s

**Table 4 sensors-18-04468-t004:** Detectability of the 4 residuals.

	$f_{kf}$	$f_{S}$	$f_{v_{v}}$	$f_{\omega_{w}}$
$R_{1}$	×	●	●	●
$R_{2}$	●	×	●	●
$R_{3}$	●	●	×	●
$R_{4}$	●	●	●	×

**Table 5 sensors-18-04468-t005:** Fault setting details of Experiment.

Fault	Type	Signal	Time Span
$f_{v_{v}}$	Bias (−10)	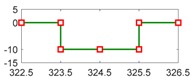	323.5–325.5 s
$f_{\omega_{w}}$	Bias (+20)	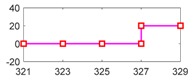	327–329 s

**Table 6 sensors-18-04468-t006:** Main parameter values of the EcoCAR2 ABS.

Parameter	Values	Unit
*K*	1	-
*m*	1897.6	kg
*g*	10	m·s^−2^
$R_{r}$	0.334	m
$K_{f}$	1	-
$T_{B}$	0.01	-
$S_{T}$	0.2	-
*I*	0.735	kg·m^2^
$v_{0}$	39.09	rad·s^−1^
$\beta_{2}$	10	-
$\beta_{3}$	30	-
$\beta_{4}$	20	-

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
