# Peer review of "Model-Based Fault Diagnosis of an Anti-Lock Braking System via Structural Analysis"

_sensors, 2018, doi:10.3390/s18124468_

Round 1

Reviewer 1 Report

This paper deals with ABS fault diagnosis based firstly on modeling the system and decomposing it in a set of known and unknown variables. Then, Strauctural Analysis is used to check the detectability and insolability of the fauls, this analysis results in a Fault signature matrix, by employing the technique of Dulmage-Mendelosohn (DM) decomposition in SA. The considered faults are: Leakage of solenoid value, The fault of slip rate operator, Internal error of ECU, Vehicle speed sensor fault, Wheel angular speed sensor fault. Then residuals are generated using the principal of Analytical Redundancy, which can be used only on the over constrained part of the system.

Although the approach used is well known in the literature, the application support considered is very interesting because it is a critical part of a vehicle. The proposed model is well explained and will interest readers. The paper is well organized and easy to understand, the diagrams are explicit. Simulation results and experimental results show the effectiveness of the proposed approach.

The introduction of paper focused on ABS and the SA method. I think that in a quality journal like Sensors, it would be necessary to present to the readers the other existing methods applied to the vehicles, like review papers on fault diagnosis, example: A survey of fault diagnosis and fault-tolerant techniques; Part I: IEEE Transactions on Industrial Electronics 62 (6) (2015) 3757-3767. Other model-based fault diagnosis techniques such as observer theory and parity space, used simultaneously in: Intelligent Monitoring of Electric Vehicles, 2009 IEEE / ASME International Conference on Advanced Intelligent Mechatronics. Singapore, July 14-17, 2009. pp. 797-804.

Reviewer 2 Report

In this paper, the authors have followed a structural analysis route for designing a fault detection system for and ABS brake system. The design is based on the developments of the system/sensing on the progression of the analysis of the performance of the system. The final system constitutes a well established design that covers all the unexpected conditions that the system might experience.

Mathematical analysis and simulations proved that the system is functional and reliable.

The paper is well presented, however, the English needs to be checked, the use of active voice should be avoided in academic writing, the use of capital letters at the beginning of a continuation paragraph need to be corrected (Where after equations)

Figure 8 - Figure 9, should be Figures 8 and 9

The sentence "By using", is not correct, please remove "By"

"Figure X shows the results ...", --> should be "Figure X shows results …"

In the conclusion, no need to define MSO, as it will not be used after this instance.

Define SA in the text, not in the abstract.
